# Molecular Identification and Expression Analysis of NOD1/2 and TBK1 in Response to Viral or Bacterial Infection in the Spotted Knifejaw (*Oplegnathus punctatus*)

**DOI:** 10.3390/ani15071006

**Published:** 2025-03-31

**Authors:** Yu Song, Lei Wang, Kaimin Li, Mengqian Zhang, Songlin Chen

**Affiliations:** 1State Key Laboratory of Mariculture Biobreeding and Sustainable Goods, Yellow Sea Fisheries Research Institute, Chinese Academy of Fishery Sciences, Qingdao 266071, China; sy302969975@163.com (Y.S.); li_kaimin2025@163.com (K.L.); mengqian154273@163.com (M.Z.); chensl@ysfri.ac.cn (S.C.); 2Laboratory for Marine Fisheries Science and Food Production Processes, Qingdao Marine Science and Technology Center, Qingdao 266237, China

**Keywords:** *Oplegnathus punctatus*, NOD1, NOD2, TBK1, immunity, infection

## Abstract

This study investigated the roles of the *Opnod1*, *Opnod2*, and *Optbk1* in the immune defense of the spotted knifejaw (*Oplegnathus punctatus*), a commercially important farmed marine fish highly susceptible to bacterial and viral infections that often cause economic losses. The *Opnod1* and *Opnod2* (pattern recognition receptor-encoding genes) and *Optbk1* (a serine/threonine kinase gene critical to immune signaling) exhibited tissue-specific expression: the *Opnod1* was predominantly expressed in the skin, the *Opnod2* in the gills, and the *Optbk1* in the liver. Following infection with the viral pathogen SKIV-SD or the bacterium *Vibrio harveyi*, these genes showed marked expression changes. Furthermore, their upregulation in kidney cells after stimulation with poly I:C (a viral mimic) and LPS (a bacterial component) underscored their responsiveness to pathogen-associated signals. These results emphasize the pivotal role of the NOD1/2-TBK1 signaling pathway in the spotted knifejaw’s immune response, offering critical insights for advancing disease-resistant aquaculture strains through targeted genetic strategies.

## 1. Introduction

Innate immunity is a key defense mechanism in vertebrates, responsible for recognizing pathogenic microorganisms and providing resistance to bacterial invasion, while also stimulating the development of acquired immunity [1].The key function of immunity lies in its ability to recognize and interact with pathogenic microorganisms [2]. When a pathogenic microorganism invades an organism, immune cells identify the pathogen or intracellular damage signals through pattern recognition receptors (PRRs) expressed on the cell membrane or in the cytoplasmic matrix [3]. Nucleotide-binding oligomerization domain (NOD)-like receptors (NLRs) are among the most widely studied pattern recognition receptors [4], which are located in the cytoplasm and involved in recognizing bacteria and viruses [5]. TBK1 is an important serine/threonine kinase that interacts with various signaling molecules and mediates the phosphorylation of IRF3 (interferon regulatory factor 3), playing a central role in regulating interferon production.

NOD-containing protein 1(NOD1) and NOD-containing protein 2(NOD2) were the first NLR family members identified and are important pattern recognition receptors for bacteria and viruses [6,7]. Both NOD1 and NOD2 contain three conserved domains: the N-terminal Caspase activating and recruitment domain (CARD), the central NOD, which mediates intermolecular oligomerization, and the C-terminal leucine-rich repeats (LRRs), which are responsible for ligand recognition [8,9]. In mammals, NOD1 specifically recognizes γ-D-Glu-mDAP (iE-DAP) in bacteria [10], while NOD2 specifically recognizes the Muramyl Dipeptide (MDP) in bacteria, triggering an immune response by activating the downstream NF-κB signaling pathway [9,11]. Upon ligand binding, the LRRs undergo conformational changes that expose the CARDs, enabling the recruitment of receptor-interacting serine-threonine kinase 2 (RIPK2) through CARD-CARD interactions. RIPK2 is subsequently polyubiquitinated by E3 ligases (e.g., XIAP, cIAP1/2), forming a scaffold that recruits the TAK1-TAB1/2 complex. TAK1 phosphorylates and activates the IKK complex (IKKα/IKKβ/NEMO), leading to the phosphorylation of the NF-κB inhibitor IκBα, which is then degraded by the proteasome. This results in the release of the NF-κB heterodimer (p50/p65), allowing its nuclear translocation to induce the transcription of pro-inflammatory cytokines (e.g., TNF-α, IL-6) and antimicrobial peptide [12,13]. In addition, NOD2 activates the mitogen-activated protein kinases (MAPK) signaling pathway, further promoting cytokine production [14]. Besides recognizing MDP, NOD2 has also been shown to recognize viral ssRNA and promote the activation of interferon regulatory factors, inducing the production of type I interferon [15]. Thus, NOD2 plays an important role in antiviral immune response by enabling cells to resist viral infection and activating acquired immunity. The *nod1* and *nod2* genes have been cloned and functionally studied in various teleost species, including rohu (*Labeo rohita*) [16], mrigal (*Cirrhinus mrigala*) [17], catla (*Catla catla*) [18], channel catfish (*Ictalurus punctatus*) [8], grass carp (*Ctenopharyngodon idella*) [19], miiuy croaker (*Miichthys Miiuy*) [20], zebrafish (*Danio rerio*) [21], and Nile tilapia (*Oreochromis niloticus*) [22]. These findings highlight the central role of NOD1/2 in vertebrate immune defense, acting as crucial hubs that integrate bacterial and viral immune signals [23,24].

TBK1 is a serine/threonine kinase commonly expressed in the IKK family, also known as NF-κB-activating kinase, which plays an important role in interferon production and antiviral innate immunity. Following viral infection, virus-specific features are recognized by PRRs, which then activate downstream signaling cascades. TBK1 is a key signaling molecule that receives signals from the TLR (Toll-like receptors), RLR (RIG-I-like receptor), and the cGAMP synthase (cGAS) pathways, and it is essential for the generation of type I IFN antiviral immune responses. In addition to its role in interferon-induced function, TBK1 has been found to participate in cellular processes such as insulin signaling, cellular autophagy and mitophagy, and antiviral and antibacterial immune responses in mammals. These findings suggest that TBK1 is not only a key molecule in the induction of interferon production but also plays an important role in regulating various cell biological processes. Homologs of TBK1 have been cloned and characterized in a range of scleractinian fish species, including zebrafish [25], Atlantic salmon (*Salmo salar*) [26], common carp (*Cyprinus carpio*) [27], grass carp [28], black carp (*Mylophyngodon piceus*) [29], crucian carp (*Carassius auratus*) [30], mrigal [26], rohu [27], catla [28], and catfish [29].

The spotted knifejaw (*Oplegnathus punctatus*) is a valuable mariculture species in China. However, the intensification of aquaculture has led to increased disease outbreaks, which have become a problem that restricts the healthy culture and promotion of spotted knifejaw. The main diseases affecting spotted knifejaw aquaculture include viral, bacterial, and parasitic infections. Recently, RSIV-type megalocytivirus (SKIV-ZJ07), SKIV-TJ, and ISKNV-type megalocytivirus (SKIV-SD) have been isolated from spotted knifejaw [31,32,33]. Bacterial diseases such as black body disease caused by *Vibrio harveyi* can lead to ascites, enteritis and severe damage to internal organs, posing a serious threat to the aquaculture industry [34]. Lipopolysaccharide (LPS), also known as endotoxin, is a component of the outer membrane of Gram-negative bacteria. LPS triggers a robust inflammatory response by activating the NF-κB pathway, leading to the release of cytokines, chemokines and other molecules that mediate inflammation. Poly I:C is a synthetic double-stranded RNA analog that mimics viral RNA, thereby triggering immune responses typically associated with viral infections. Both LPS and Poly I:C act as pathogen-associated molecular patterns recognized by PRRs, which initiate immune reactions that often mimic bacterial and viral infections, respectively [24,35,36].

Therefore, in this study, the coding sequence (CDS) regions of the *Opnod1*, *Opnod2,* and *Optbk1* genes were cloned, and their expression patterns in various tissues of spotted knifejaw were examined using quantitative real-time PCR (qRT-PCR). Additionally, the expression levels of these genes were analyzed in different tissues following infection with SKIV-SD or *V. harveyi*. The goal was to further investigate the role of these genes in the immune response of spotted knifejaw and provide a foundation for further studies on their involvement in the immune response.

## 2. Materials and Methods

### 2.1. Experimental Fish

The spotted knifejaw used in this study were obtained from Mingbo Aquaculture Co., Ltd., Yantai, Shandong, China. The average body weight of these fish is about 150 g, and they were healthy without disease. Before the experiment, the fish were temporarily kept in a water tank for one week, with the water temperature maintained at about 25 °C.

### 2.2. Sample Processing and Collection

Eleven tissues of liver, spleen, kidney, head kidney, heart, intestine, gills, brain, skin, muscle, and blood were collected from five randomly selected spotted knifejaws after being anesthetized by MS-222. The tissues were rapidly placed in liquid nitrogen and then transferred to a −80 °C refrigerator for further processing.

For the spotted knifejaw SKIV-SD infection experiment, 20 healthy spotted knifejaw were selected. The virus used in this experiment was provided by Professor Qiwei Qin from South China Agricultural University. The individuals were anesthetized and injected intraperitoneally with 100 µL (10^9^ copies) of virus solution per fish. Samples were collected at five time points: 0, 1, 4, 7, and 10 d. Three individuals were randomly selected at each time point, and three tissues (liver, spleen, and kidney) were taken from each fish. Tissues were rapidly frozen in pre-prepared liquid nitrogen and stored in a −80 °C refrigerator for subsequent RNA extraction.

For the *V. harveyi* infection experiment, 50 healthy spotted knifejaw were selected. The *V. harveyi* strain used in the experiment were kept in our laboratory. The *V. harveyi* solution was diluted to 1 × 10^9^ cfu/mL with PBS, and the spotted knifejaw were anesthetized and intraperitoneally injected with 100 µL of the bacterial solution per fish. Samples were taken at eight time points: 0, 6, 12, 24, 48, 72, 96, and 120 h. Three spotted knifejaw were sampled at each time point, and three tissues (liver, spleen, and kidney) were taken from each fish. The tissues were immediately frozen in liquid nitrogen and promptly stored in a −80 °C refrigerator for subsequent experiments.

### 2.3. Total RNA Extraction and cDNA Synthesis

Total RNA was extracted from each tissue using an RNA extraction kit (Invitrogen, Carlsbad, CA, USA). The integrity of the RNA was verified by 1% agarose gel electrophoresis, and the concentration and purity of the RNA was measured by spectrophotometer. The cDNA was synthesized using the Prime Script RT reagent kit with gDNA eraser (Takara Bio, Kusatsu, Japan).

### 2.4. Amplification of the CDS Region of the Opnod1, Opnod2 and Optbk1 Genes

Based on the sequence information of the *Opnod1*, *Opnod2,* and *Optbk1* genes, primers were designed for common PCR amplification using the newly synthesized cDNA as template to verify the integrity of their open reading frame (ORF) regions. The PCR reaction system comprised 25 μL KOD^TM^ OneMaster Mix (Toyobo, Osaka, Japan), 1 μL ORF-specific primer pair (F/R), and 2 μL cDNA template, and the PCR amplification conditions were as follows: 98 °C for 5 min; 98 °C for 10 s, 59 °C for 5 s, 72 °C for 30 s, 35 cycles; 72 °C for 7 min, then stored at 4 °C. PCR products were analyzed by agarose gel electrophoresis, and target fragments of the expected size were excised from the gel and purified using a DNA gel extraction kit (Vazyme, Nanjing, China). The recovered product was ligated to pEASY-E1 vector (TransGen Biotech, Beijing, China) and transformed into Trans-T1 receptor cells (TransGen Biotech, Beijing, China), which were coated and cultured overnight. Finally, the positive monoclonal clones were selected and sent to Qingdao Ruibiotech Company (Qingdao, China) for sequencing to obtain the ORF region sequences of the *Opnod1*, *Opnod2,* and *Optbk1*.

### 2.5. Sequence Analysis of the Opnod1, Opnod2 and Optbk1 Genes

The *Opnod1*, *Opnod2,* and *Optbk1* gene sequences were analyzed using Editseq 77.1 software to predict their amino acids sequences, molecular weights, and isoelectric points. Signal peptides were predicted using an online software (http://www.cbs.dtu.dk/services/SignalP/, accessed on 21 September 2024) while transmembrane regions were analyzed using the online website (http://www.cbs.dtu.dk/services/TMHMM/, accessed on 25 September 2024). The domain prediction was performed using the online software (http://smart.embl-heidelberg.de/, accessed on 2 October 2024). Homology comparison of amino acids was carried out using DNAMAN 8 software and evolutionary trees were constructed by MEGA X 10.0.4 software.

### 2.6. Detection of Expression Patterns of the Opnod1, Opnod2 and Optbk1 Genes

qRT-PCR was applied to detect the relative expression levels of the *Opnod1*, *Opnod2,* and *Optbk1* genes in different tissues of healthy spotted knifejaw, as well as in immune-related tissues after stimulation with SKIV-SD or *V. harveyi*. β-actin was selected as the internal reference gene. The primers used for qRT-PCR are listed in Table 1. Quantification of the *Opnod1*, *Opnod2* and *Optbk1* genes was performed on an ABI 7500 Fast Real-time instrument (Applied Biosystems, Carlsbad, CA, USA) using the SYBR^®^ Premix Ex Taq^TM^ kit (Takara Bio, Kusatsu, Japan) according to the instructions. Each sample was tested in triplicate, and the relative expression of genes was calculated using the 2^−ΔΔCt^ method. All experimental data were expressed as mean ± standard error (mean ± SE) and subjected to one-way ANOVA and Duncan’s multiple comparisons using SPSS16.0 software. Differences were considered significant at *p* < 0.05 and highly significant when *p* < 0.01.

### 2.7. In Vitro Stimulation of Kidney Cells of Spotted Knifejaw with Poly I:C and LPS

The monolayer cultured was a well-grown spotted knifejaw kidney cell line, which was inoculated into a 12-well plate and left for 24 h until the cell coverage reached about 90%. The medium of the 12-well plate was aspirated off and washed three times with 1 × PBS, followed by replacement with fresh L15 medium. Spotted knifejaw kidney cells were stimulated with poly I:C at final concentrations of 0, 50, 100, and 200 μg/mL in 12-well plates, while an equal volume of PBS was added to the control wells. LPS at final concentrations of 0, 10, 50, and 100 μg/mL was added to 12-well plates, with an equal volume of PBS added to the control group. The above cell samples were cultured in a 24 °C incubator for 6 h. The cell samples were collected using Trizol and stored in a −80 °C refrigerator for use in the subsequent RNA extraction, following the procedure described in Section 2.6.

## 3. Results

### 3.1. Sequence Characteristics of the Opnod1, Opnod2 and Optbk1 Genes

As shown in Figure 1, the ORF region of the *Opnod1* gene was 2757 bp in length and encodes a 918 amino acid (aa) protein with a predicted molecular weight of 103.23 kDa and a theoretical isoelectric point of 6.447. Analysis of the OpNOD1amino acids sequence revealed the absence of a signal peptide or transmembrane region. Domain prediction indicated that OpNOD1 contained a CARD (12–94 aa), a NOD (187–360 aa), and seven C-terminal LRR domains (690–717 aa, 743–910 aa). As shown in Figure 2, the ORF region of the *Opnod2* gene was 2970 bp in length encoding a 989 aa protein with a predicted molecular weight of 110 kDa and a theoretical isoelectric point of 6.552 (Figure 3). This protein also lacked a signal peptide or transmembrane region. Additionally, the domain prediction revealed that the OpNOD2 protein has two CARDs (1–91 aa and 112–206 aa), a NOD (274–444 aa), and six C-terminal LRR domains. The ORF region of the *Optbk1* gene was 2172 bp, encoding 723 amino acids, with a molecular weight of 83.0 kDa, and a theoretical isoelectric point of 4.97. Predictions using SMART online software (http://smart.embl-heidelberg.de/, accessed on 2 October 2024) showed that the OpTBK1 protein has an S_TKc structural domain spanning amino acids positions 9 to 306 (Figure 3).

### 3.2. Amino Acids Multiple Sequence Alignment and Phylogenetic Tree Analysis

The BLAST tool (https://blast.ncbi.nlm.nih.gov/Blast.cgi, accessed on 10 October 2024) comparison revealed that the amino acids of the OpNOD1 and OpNOD2 showed high homology with NOD1 and NOD2 from other teleosts. As shown in Table 2, the similarity between the OpNOD1 and Nile tilapia NOD1 was 86.27%, with large yellow croaker (*Larimichthys crocea*) and Japanese flounder (*Paralichthys olivaceus*) showing similarities of 85.07% and 84.39%, respectively. The similarity with mouse (*Mus musculus*) and human (*Homo sapiens*) was 50.33% and 49.6%, respectively. The OpNOD2 shows 89.59% similarity with Asian Seabass (*Lates calcarifer*), 83.22% with Japanese flounder (*Paralichthys olivaceus),* and 79.27% with half-smooth tongue sole (*Cynoglossus semilaevis)*. The similarities of OpNOD2 with mouse and human were 45.54% and 46.26%, respectively.

A multiple sequence alignment of vertebrate TBK1 amino acids was performed using DNAman, revealing a high degree of conservation among different species. A phylogenetic tree of vertebrates, constructed by the Neighbor-joining method using MAGE 7 software, showed that TBK1 from mouse, human, and chimpanzee (*Pan troglodytes*) clustered together. The African clawed toad (*Xenopus laevis*) formed a separate branch, while the spotted knifejaw TBK1 formed a distinct clade with other teleost species. This suggests that spotted knifejaw TBK1 is evolutionarily more distantly related to mammals, but more closely related to other bony fishes such as striped knifejaw and semi-smooth tongue sole.

As shown in Appendix A, the amino acid sequences of NOD1, NOD2, and TBK1 are relatively conserved across species, with the conserved domains showing the highest degree of conservation. A phylogenetic tree of NOD1 and NOD2 was constructed by the Neighbor-joining method by MEGA X 10.0.4 software. Phylogenetic analysis revealed that the OpNOD1 and OpNOD2 protein clustered into a single clade with other teleosts, while the NOD1 and NOD2 of mammals were gathered in another clade. The OpNOD1 is evolutionarily most closely related to large yellow croaker, while The OpNOD2 is closely related to Asian seabass (Figure 4). Mouse (*Mus musculus*), human (*Homo sapiens*), and chimpanzee (*pan troglodytes*) TBK1 clustered together. The African clawed frog (*Xenopus laevis*) formed a distinct evolutionary branch. OpTBK1 clustered with other bony fishes into a separate group. This suggests that OpTBK1 is more distantly related to mammals and more closely related to barred knifejaw (*Oplegnathus fasciatus*) and other bony fishes such as the half-smooth tongue sole (Figure 5).

### 3.3. Expression Patterns of the Opnod1, Opnod2 and Optbk1 Genes in Healthy Individuals

The expression of the *Opnod1* and *Opnod2* genes was analyzed across 11 tissues in healthy individuals. The *Opnod1* gene exhibited high expression in the skin and intestine, while showing relatively uniform expression levels in the liver, spleen, gill, and heart. Expression was relatively low in other tissues (Figure 6A). The *Opnod2* gene showed the highest relative expression in the gill, blood, and skin, followed by the kidney, intestine, spleen, head kidney, heart, muscle, and brain (Figure 6B). The *Optbk1* gene was detected in all tissues, with the highest expression level in the liver and relatively high expression in the brain, gills, head kidney, skin, heart, and stomach (Figure 6C).

### 3.4. Changes in the Opnod1, Opnod2 and Optbk1 Gene Expression After Iridovirus Stimulation

The expression changes in the *Opnod1* and *Opnod2* at different time points in three tissues after infection with SKIV-SD are shown in Figure 7(A1–A3). Compared with the control group at day 0, the relative expression levels of the *Opnod1* gene in the liver showed no significant changes at days 1 and 4 but increased significantly at day 7, reaching approximately 22.5 times higher than that at day 0 (*p* < 0.01), before decreasing at day 10. In the spleen, the *Opnod1* expression exhibited an overall up-regulation trend, peaking at day 10, 3.8 times greater than the level at day 0. In the kidney, the *Opnod1* expression was significantly up-regulated at day 7 (*p* < 0.01), reaching 13.8 times the level at day 0, and returned to normal levels by day 10.

As shown in Figure 7(B1–B3), the *Opnod2* in the liver showed no significant changes compared with the control group at days 1 and 4 but increased significantly at day 7 (*p* < 0.01) before returning to normal levels at day 10. In the spleen, the *Opnod2* expression displayed an overall increasing trend, peaking at day 4; in the kidney, the *Opnod2* expression significantly increased, reaching its peak at day 7 (*p* < 0.01) and subsequently returning to normal levels.

After intraperitoneal injection of iridovirus, *Optbk1* expression was up-regulated in all three immunized tissues, though the expression patterns varied across tissues. In the liver, *Optbk1* expression was significantly up-regulated at days 1 and 4. In the spleen, an increase was observed at day 4 post-injection. In the kidney, Optbk1 expression was significantly elevated at days 1, 7 and 10, with the highest expression level occurring at day 1 post-injection, reaching 4.3 times the level of the control group (Figure 7(C1–C3)).

### 3.5. Changes in the Expression of the Opnod1, Opnod2 and Optbk1 Genes After V. harveyi Stimulation

After *V. harveyi* infection, the *Opnod1* gene expression levels were significantly up-regulated in the spleen at 12 h post infection, whereas in the liver, a significant increase was observed at 72 h. In the kidney, the relative expression level of the *Opnod1* was significantly up-regulated at 48 h, with the peak expression occurring later than in the spleen (Figure 8(A1–A3)).

The expression pattern of the *Opnod2* was similar to that of the *Opnod1*. In the liver, the *Opnod2* gene expression levels were significantly up-regulated at 72 h post injection. In the spleen, the expression pattern of *Opnod2* gene reached relatively high levels at 12 h and then gradually declined. However, in the kidney, the relative expression showed no significant changes (Figure 8(B1–B3)).

Expression of Optbk1 in all three tissues was significantly up-regulated compared with the control group at 12 h post injection, showing 2.2, 1.8, and 1.0-fold increases in the liver, spleen, and kidney, respectively. Additionally, *Optbk1* expression in the liver and spleen remained significantly up-regulated at 72 h, while in the kidney, significant up-regulation was observed at 0 h and 24 h post stimulation (Figure 8(C1–C3)).

### 3.6. In Vitro Stimulation of Grouper Kidney Cells

Spotted knifejaw kidney cells were stimulated with different concentrations of poly I:C and LPS for 6 h. qRT-PCR was performed to assess changes in expression of *Opnod1*, *Opnod2,* and *Optbk1*. The expression of *Opnod1* and *Optbk1* was upregulated 2–3 fold at poly I:C concentrations of 10–100 μg/mL, while the *Opnod2* expression in kidney cells increased 60–80 fold with poly I:C stimulation (Figure 9A–C). After the stimulation with LPS, the expression levels of the *Opnod1*, *Opnod2,* and *Optbk1* exhibited a 1.5–4 fold increase at 50 μg/mL, followed by a decrease at 100 μg/mL (Figure 9D–F).

## 4. Discussion

In this study, the full length CDS regions of the *Opnod1* and *Opnod2* genes were cloned and then subjected to sequence homology and evolutionary analysis with the NOD1 and NOD2 from other species. The results revealed that the *Opnod1* and *Opnod2* contained all the characteristic domains of the NLR family [13]. The protein domains of the OpNOD1 and OpNOD2 are highly conserved compared to those of other species, suggesting that they likely share similar functional properties in pathogen recognition. The OpNOD1 contains seven LRR domains, similar to those found in Nile tilapia [22] and orange-spotted grouper (*Epinephelus coioides*) [36], while NOD1 genes of grass carp, channel catfish, and zebrafish have six LRR domains [19,37]. The C-terminus of the OpNOD2 has six LRR domains, which are identical to those observed in the miiuy croaker [20] and orange-spotted grouper [36]. However, grass carp and zebrafish have five LRRs. NOD1 and NOD2 distinguish different ligands through their LRRs domains, enabling the immune system to respond effectively to invading pathogens.

Species such as the spotted knifejaw, which inhabit marine environments with high pathogen loads, are often exposed to substantial pathogen challenges. As a result, their immune receptors tend to have more LRR domains, allowing for the recognition and response to a wider variety of pathogens. This observation is consistent with findings from studies on other species [4,38].

We found that the *Opnod1* and *Opnod2* were widely expressed in various tissues, as detected using qRT-PCR. The *Opnod1* gene was highly expressed in the skin, similar to findings in the orange-spotted grouper. The high expression of the *Opnod1* in the skin can be attributed to its role as the body’s first line of defense, requiring efficient pathogen surveillance and a rapid immune response, particularly against bacteria. Through its pattern recognition function, the *Opnod1* aids the skin in detecting pathogens, regulating immune responses and promoting wound healing, thus playing a crucial role in the skin’s immunity. In contrast, in healthy zebrafish, grass carp, and channel catfish, the relative expression of NOD1 was highest in the spleen and lowest in the blood; the *Opnod2* gene was highly expressed in gill, blood, and skin of spotted knifejaw, reflecting an adaptive response of the fish immune system. These tissues play crucial roles in immune defense, particularly in recognizing and eliminating external pathogens. The gills are exposed to microorganisms in the water, the blood transports immune cells that monitor the entire body, and the skin acts as an external barrier, directly interacting with pathogens. The elevated expression of the *Opnod2* in these tissues enhances the immune response, offering protection from pathogen invasion. In Nile tilapia, the *nod2* gene was highly expressed in the spleen, while in orange-spotted grouper, the *nod2* gene was most abundant in the kidney, followed by the skin and gill. This pattern is similar to that observed in the spotted knifejaw. Interestingly, both the *nod1* and *the nod2* genes show the highest expression levels in the head kidney of Chinese perch (*Siniperca chuatsi*) [39].

Taken together, these findings suggest that the *Opnod1* and *Opnod2* have an important role in the innate immunity of spotted knifejaw. The involvement of NOD1 and NOD2 in antibacterial and antiviral immune responses has been reported in a few species of fish, including the Indian major carp (*Labeo rohita*) [40], mrigal [41], orange-spotted grouper [36], rainbow trout (*Oncorhynchus mykiss*) [11], grass carp [19], channel catfish [8], and zebrafish [21]. In Nile tilapia, significant changes in the expression levels of NOD1 and NOD2 were observed in the blood, spleen, kidney, intestine, and gills following an injection of *S.agalactiae* [22]. Similarly, in mrigal, the expression levels of NOD1 and NOD2 in the gills, liver, kidney, and intestine significantly increased after infection with either *S.agalactiae* or *Aeromonas hydrophila* [41]. In this study, we observed significant changes in the relative expression levels of the *Opnod1* and *Opnod2* genes in the liver, spleen, and kidney tissues of spotted knifejaw after infection by SKIV-SD or *V. harveyi*, further confirming the crucial role of the *Opnod1* and *Opnod2* in the immune response of the spotted knifejaw.

Studies in mammals have shown that NOD1 protein recognizes the peptidoglycan molecules of iE-DAP to detect bacterial pathogens. Mammalian NOD2 detects MDP in peptidoglycans of both Gram-positive and Gram-negative bacteria, and both receptors activate immune responses through activation of the NF-κB signaling pathway [42]. However, the specific ligands of NOD1 and NOD2 in teleost fish remain unclear. In zebrafish, NOD2 responded significantly to MDP stimulation, but not to Poly I: C or LPS [21], suggesting that zebrafish NOD2 may function as an MDP receptor, which is consistent with Nile tilapia [22]. In our research, the *Opnod1* and *Opnod2* were found to play crucial roles in antiviral and antibacterial infection processes.

In mammals, TBK1 has been widely studied as an important molecular bridge connecting the signaling of TLRs (such as TLR3 and TLR4) and RLRs. This activation triggers transcription factors IRF3 and IRF7, leading to the production of type I interferons. In this study, we identified the *Optbk1* gene. After multiple sequence comparisons we found that the TBK1 sequence was highly conserved, and phylogenetic analysis of TBK1 showed that *Optbk1* was more closely related to *tbk1* from other teleost species, suggesting it may be a homolog of mammalian *tbk1*. Tissue expression analysis showed that *Optbk1* was expressed in the liver and brain, highlighting its crucial roles in immune defense, autophagy, and neuroprotection. In the liver, *Optbk1* activates antiviral responses and clears damaged cells, ensuring the proper functioning of the immune system. In the brain, it regulates neuro immune responses and autophagy processes, protecting neurons from injury. The elevated expression of *Optbk1* enhances the fish’s ability to cope with environmental stress, pathogen infections, and metabolic challenges, thereby supporting overall survival and health. The expression pattern of TBK1 was slightly different across species. In grass carp, TBK1 was mainly expressed in the spleen [43], while in the large yellow croaker, TBK1 was highly expressed in the brain [44]. In spotted rock sea bream, the highest TBK1 expression level was found in the liver, with lower expression in the spleen and the head kidneys. The pattern was similar to the tissue expression of TBK1 in the dark sandpiper (*Odontobutis obscurus*) [45], suggesting that OpTBK1 may not be required in the immune tissues of healthy spotted knifejaw. In black carp, TBK1 has been shown to exhibit antiviral activity against grass carp SVCV and GCRV [29], while in the large yellow croaker, TBK1 can interact with the E3 ubiquitin ligase, Nrdp1, to help defend against infections [44].

## 5. Conclusions

In this study, the expression of the *Opnod1*, *Opnod2,* and *Optbk1* was up-regulated to different degrees in immune tissues liver, spleen, and kidney after stimulation by SKIV-SD and *V. harveyi*. Additionally, spotted knifejaw kidney cells showed up-regulation to poly I:C and LPS in vitro. These results suggest that the NOD1/2-TBK1 signaling pathway plays a crucial role in the immune response of teleosts, but its molecular mechanism remains poorly understood.

In summary, this study showed that the *Opnod1*, *Opnod2,* and *Optbk1* are involved in the immune response process of spotted knifejaw, providing valuable evidence for further research on the role of the NOD1/2-TBK1 signal pathway in the immunity of spotted knifejaw against diseases.

## Figures and Tables

**Figure 1 animals-15-01006-f001:**
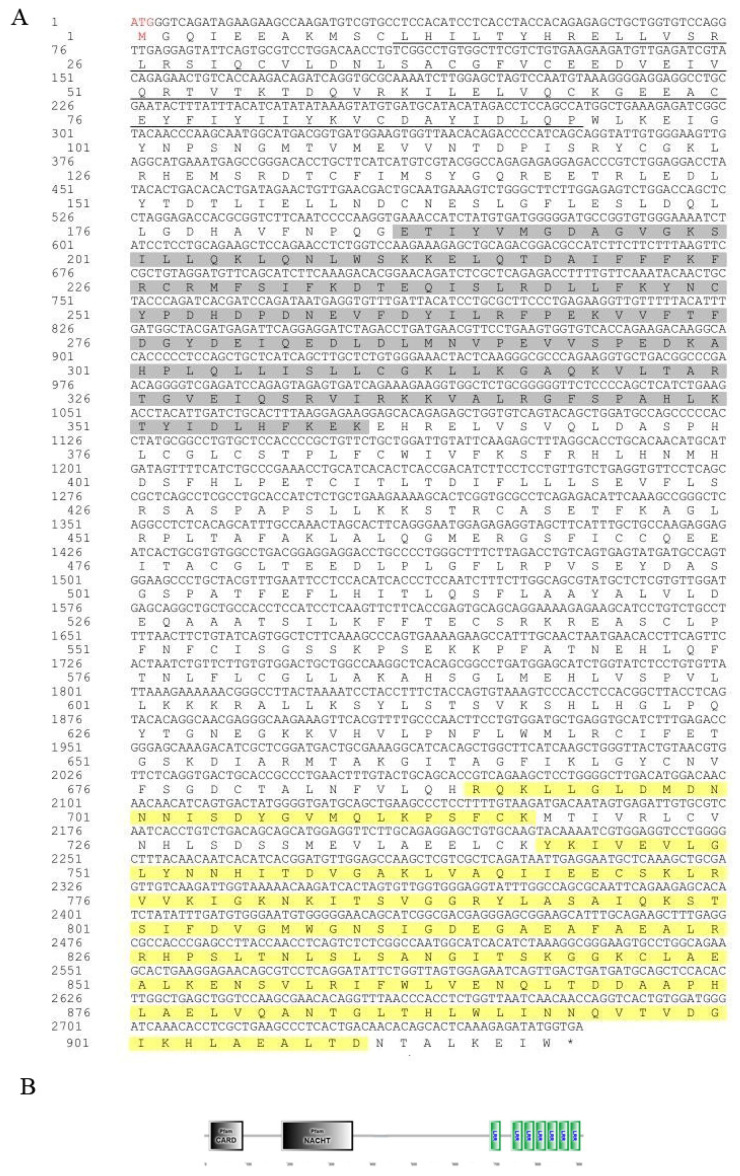
Amino acids sequence and predicted domains of the OpNOD1 protein. (**A**) The translated amino acid sequence is shown below the nucleotide sequence. The start codon “ATG” is marked in red, and the stop codon “TAG” is indicated with an asterisk. The CARD is underlined, the NOD is shaded in gray, and the LRR domains are highlighted in yellow. (**B**) The protein domain prediction of OpNOD1. The CARD and NACHT domain are shaded in gray, and the LRR domains are marked in green.

**Figure 2 animals-15-01006-f002:**
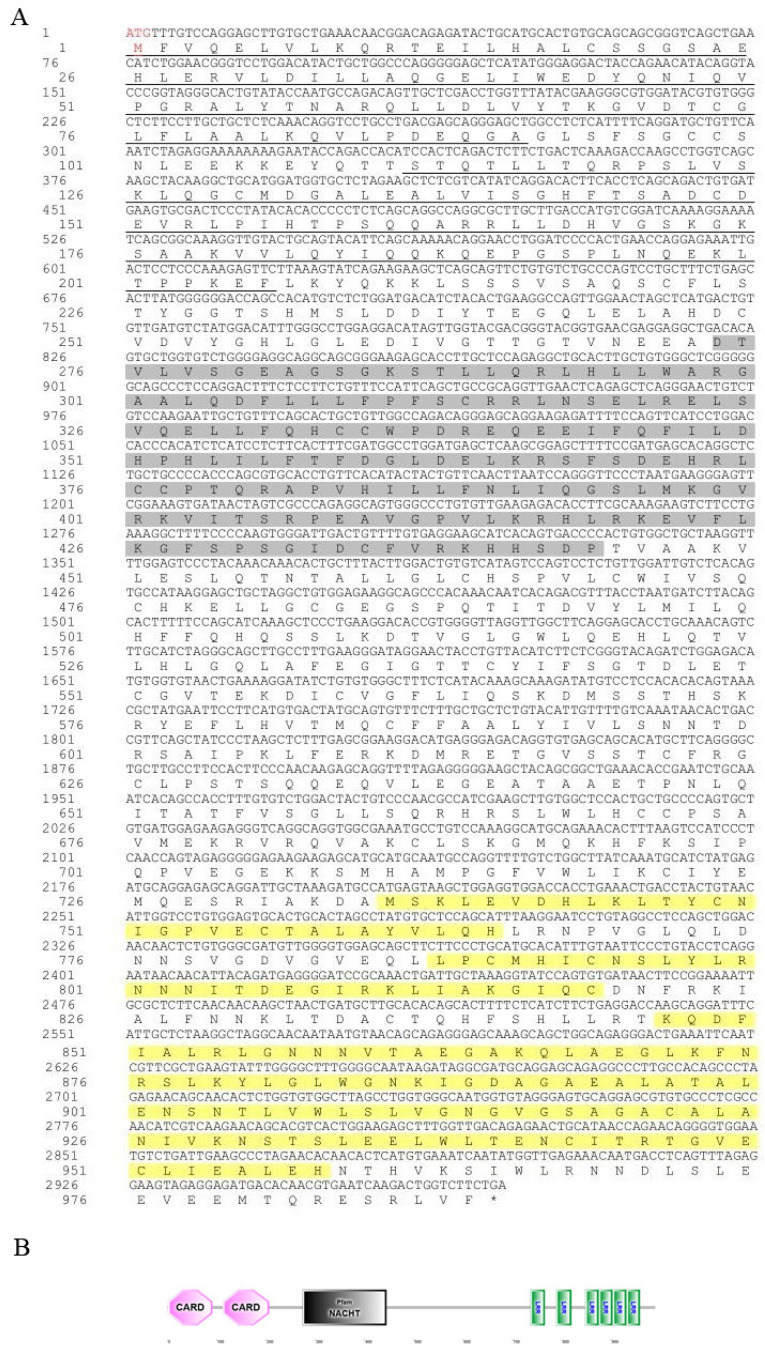
Amino acid sequence and predicted domains of the OpNOD2 protein. (**A**) The translated amino acid sequence is shown below the nucleotide sequence. The start codon “ATG” is marked in red, and the stop codon “TGA” is indicated with an asterisk. (**B**) The protein domain prediction of OpNOD2. The CARD is underlined, the NACHT domain is shaded in gray, and the LRR domains are marked in green.

**Figure 3 animals-15-01006-f003:**
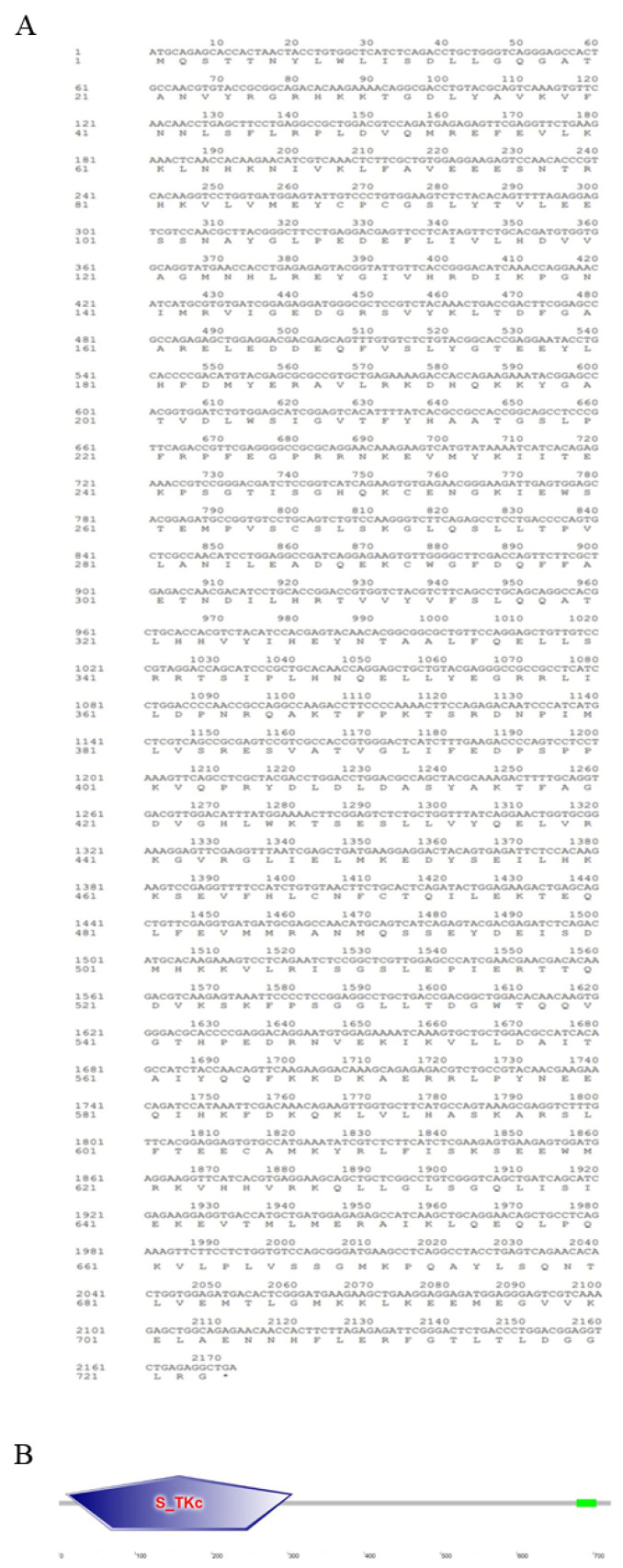
Amino acid sequence and predicted domains of the OpTBK1 protein. (**A**) The translated amino acid sequence is shown below the nucleotide sequence. (**B**) The protein domain prediction of OpTBK1. The S-TKc domain is marked with an asterisk.

**Figure 4 animals-15-01006-f004:**
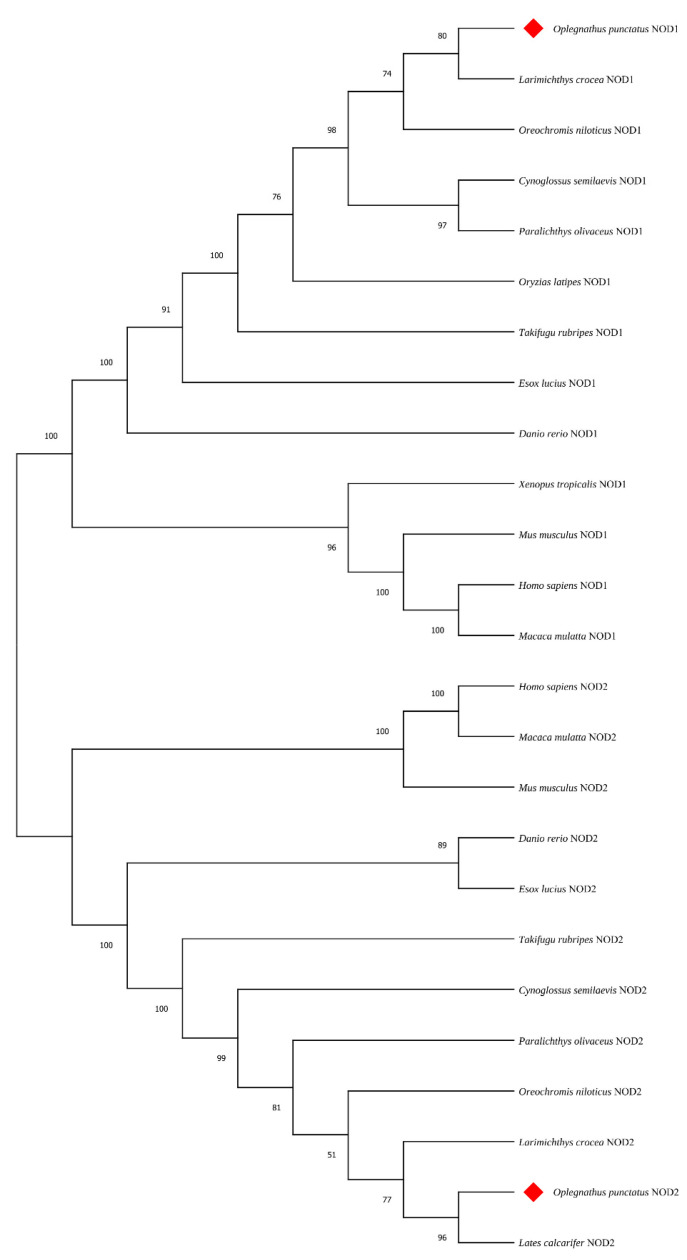
Phylogenetic tree based on the amino acid sequences of NOD1 and NOD2 from various species. Numbers at nodes represent NJ bootstrap values. *O. punctatus* is highlighted in red diamond to indicate its evolutionary position.

**Figure 5 animals-15-01006-f005:**
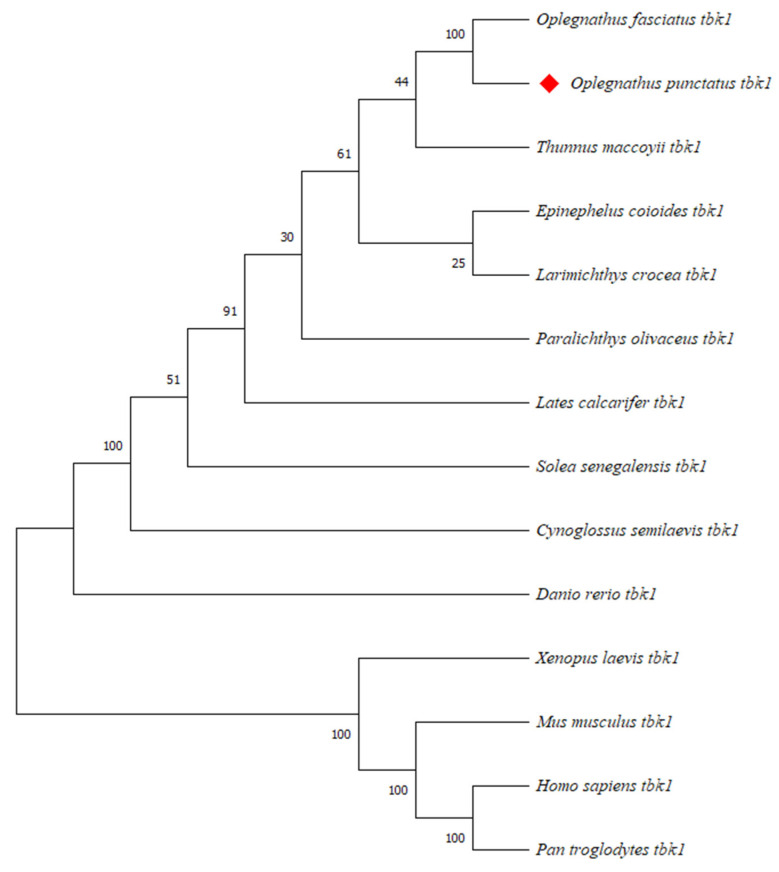
Phylogenetic tree based on the amino acid sequences of TBK1 from various species. Numbers at nodes represent NJ bootstrap values. *O. punctatus* is highlighted in red diamond to indicate its evolutionary position.

**Figure 6 animals-15-01006-f006:**
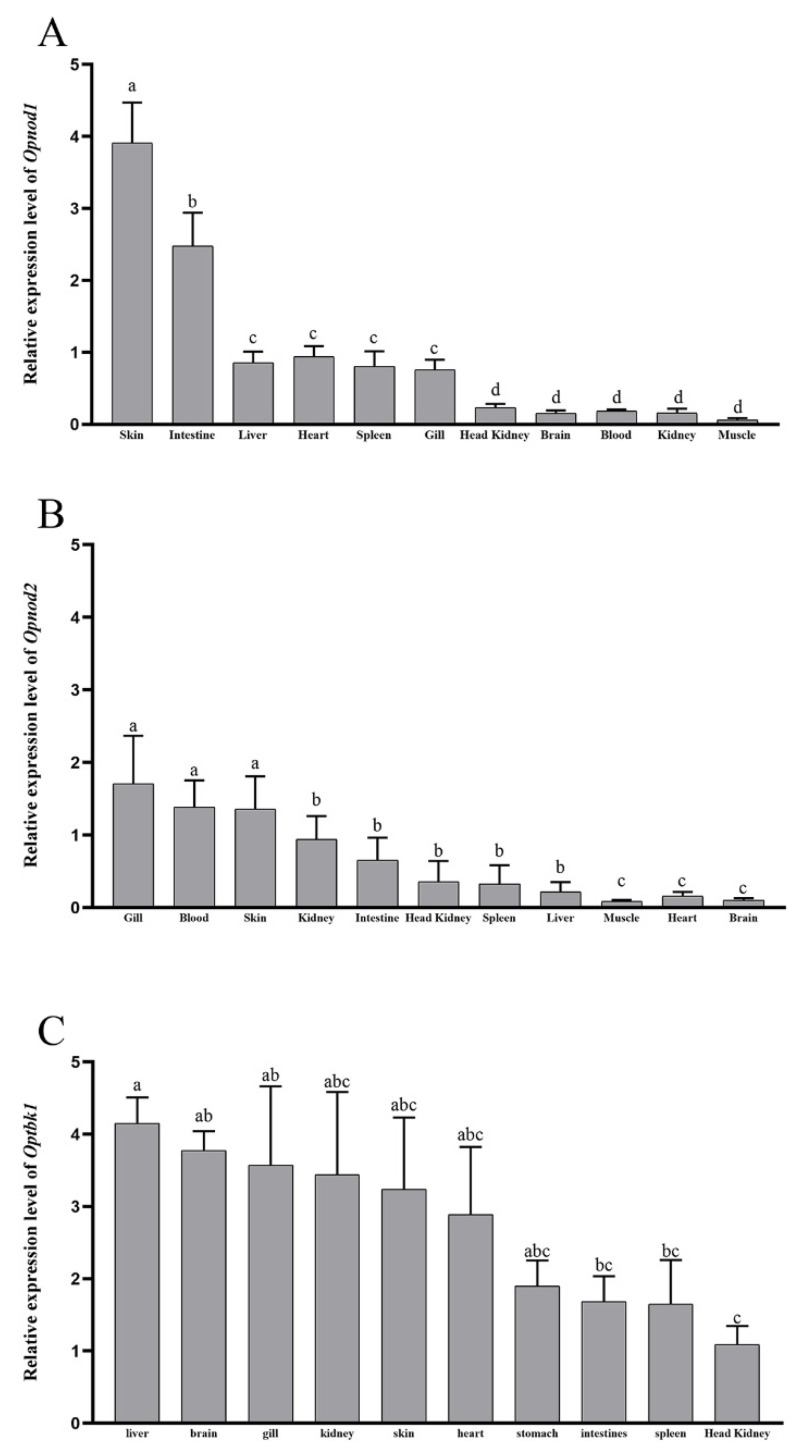
Expression of the *Opnod1* (**A**)*, Opnod2* (**B**), and *Optbk1* (**C**) mRNA in different tissues of healthy spotted knifejaw. All the data are shown as mean ± SE (*n* = 5). The different letters represent significant differences.

**Figure 7 animals-15-01006-f007:**
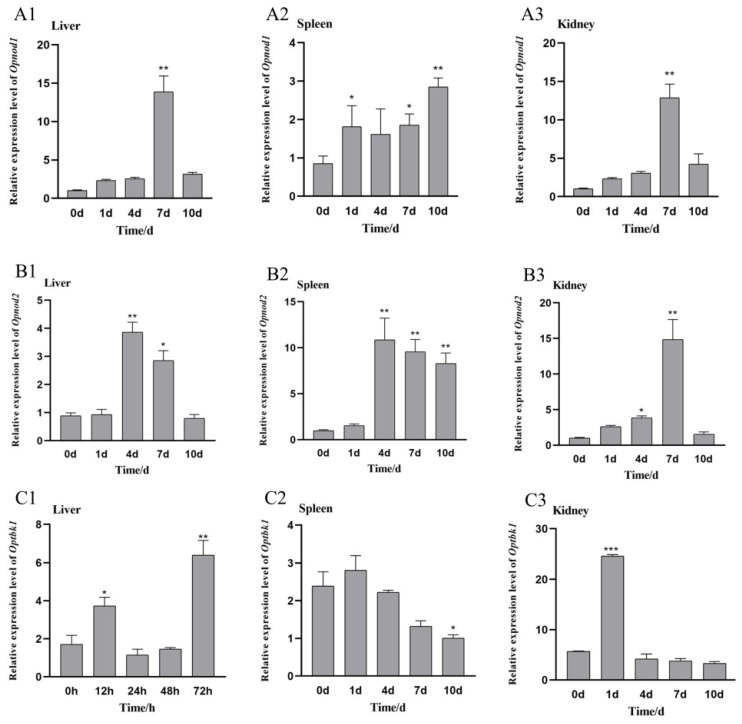
Relative expression of the *Opnod1* (**A**), *Opnod2* (**B**), and *Optbk1* (**C**) genes in the liver, spleen, and kidney of spotted knifejaw at different time points after SKIV-SD infection. All the data are shown as mean ± SE (*n* = 5). The asterisk indicates significant differences (* *p* < 0.05, ** *p* < 0.01, *** *p* < 0.001).

**Figure 8 animals-15-01006-f008:**
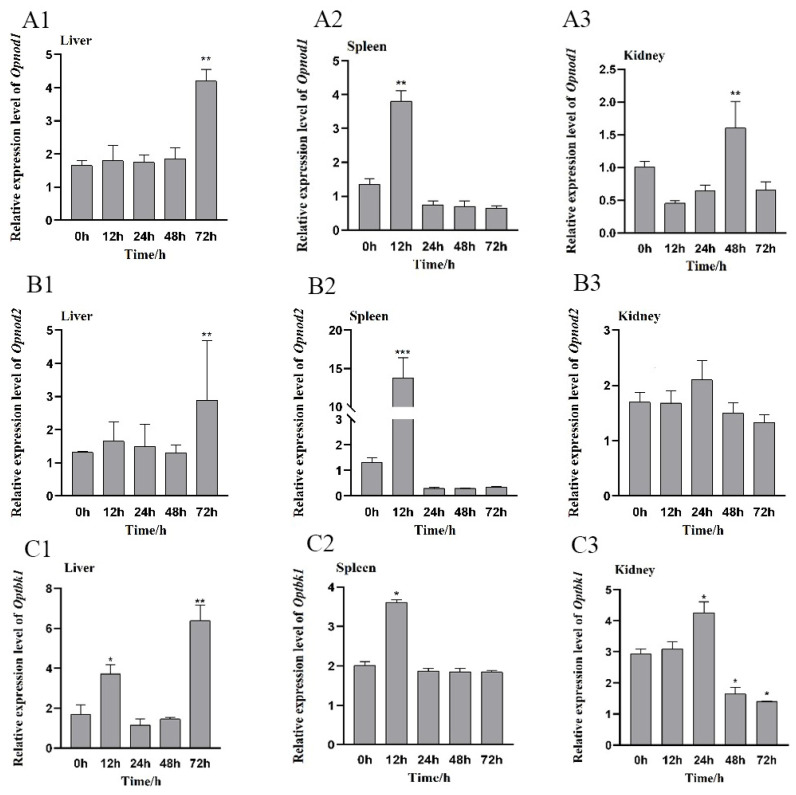
Relative expression levels of the *Opnod1* (**A**), *Opnod2* (**B**), and *Optbk1* (**C**) genes in the liver, spleen and kidney of spotted knifejaw at different time points after *V. harveyi* infection. All the data are shown as mean ± SE (*n* = 5). The asterisk indicates significant differences (* *p* < 0.05, ** *p* < 0.01, *** *p* < 0.001).

**Figure 9 animals-15-01006-f009:**
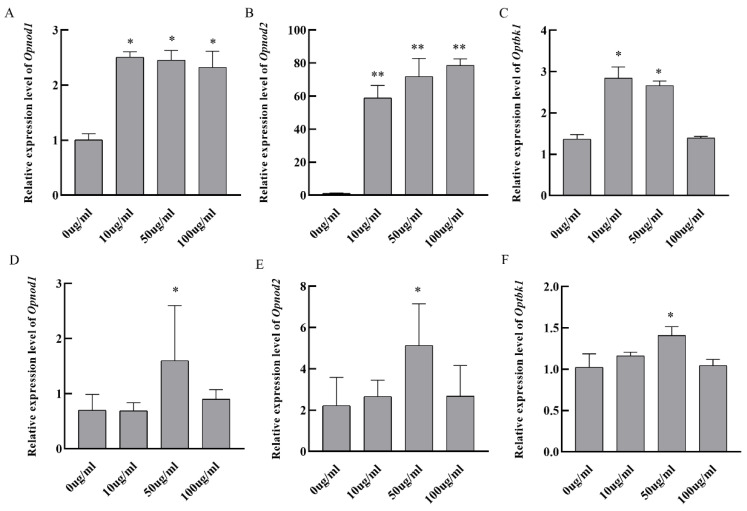
Expression levels of the *Opnod1*, *Opnod2,* and *Optbk1* in kidney cells stimulated with different concentrations of poly I:C and LPS. (**A**–**C**): expression levels in kidney cells treated with different concentrations of poly I:C; (**D**–**F**): expression levels in kidney cells treated with different concentrations of LPS. The asterisk indicates significant differences (* *p* < 0.05, ** *p* < 0.01).

**Table 1 animals-15-01006-t001:** Primers used in this study.

Primer	Sequence(5′-3′)	Use Application
Opnod1-ORF-F	ATGGGTCAGATAGAAGAAGCCAAG	ORF verification
Opnod1-ORF-R	TCACCATATCTCTTTGAGTGCTGTG	
Opnod2-ORF-F	ATGTTTGTCCAGGAGCTTGTGCTG	
Opnod2-ORF-R	TCAGAAGACCAGTCTTGATTCACG	
Optbk1-ORF-F	ATGCAGAGCACCACTAACTACCTG	
Optbk1-ORF-R	TCAGCCTCTCAGACCTCCGTCCAG	
Opnod1-qRT-F	GTTGGTGGGAGGTATTTGG	qRT-PCR
Opnod1-qRT-R	GTTGGTAAGGCTCGGGTG	
Opnod2-qRT-F	GGGGCAATAAGATAGGCG	
Opnod2-qRT-R	TGACGATGTTGGCGAGGG	
Optbk1-qRT-F	AGGACGACGAGCACTTTGTG	
Optbk1-qRT-R	CGTATTTCTTCTGGTGGTCTTTT	
β-actin-F	GCTGTGCTGTCCCTGT	
β-actin-R	GAGTAGCCACGCTCTGTC	

**Table 2 animals-15-01006-t002:** Amino acids similarity of NOD1, NOD2, and TBK1 proteins among *Oplegnathus punctatus* and other vertebrates.

	Species	GenBank Access Number	Similarity/%
NOD 1	*Oreochromis niloticus*	XP_005472430.1	86.27
	*Larimichthys crocea*	XP_019134818.2	85.07
	*Paralichthys olivaceus*	XP_019946646.1	84.39
	*Cynoglossus semilaevis*	XP_008322367.1	81.05
	*Oryzias latipes*	XP_020565632.1	78.45
	*Takifugu rubripes*	XP_003965935.3	74.4
	*Esox lucius*	XP_010883447.1	71.35
	*Danio rerio*	XP_002665106.3	65.25
	*Macaca mulatta*	XP_028701734.1	50.71
	*Mus musculus*	NP_001164478.1	50.33
	*Homo sapiens*	XP_011513383.1	49.6
	*Xenopus laevis*	XP_031759856.1	48.74
NOD2	*Lates calcarifer*	XP_018522174	89.59
	*Larimichthys crocea*	XP_010727419.3	85.64
	*Oreochromis niloticus*	XP_003437591.1	83.82
	*Paralichthys olivaceus*	XP_019935411.1	83.22
	*Cynoglossus semilaevis*	XP_008335431.1	79.27
	*Takifugu rubripes*	XP_029701512.1	77.25
	*Esox lucius*	XP_010894874.4	67.4
	*Danio rerio*	NP_001314973.1	64.18
	*Macaca mulatta*	XP_014981593.2	46.26
	*Homo sapiens*	NP_071445.1	46.26
	*Mus musculus*	AAN84594.1	45.54
TBK1	*Oplegnathus fasciatus*	AHX37216.1	99.86
	*Thunnus maccoyii*	XP_042259009.1	98.47
	*Larimichthys crocea*	AKM77645.1	98.06
	*Epinephelus coioides*	ATI15615.1	97.93
	*Lates calcarifer*	XP_018530412.1	97.92
	*Paralichthys olivaceus*	XP_019966450.1	97.09
	*Solea senegalensis*	XP_043878151.1	96.26
	*Cynoglossus semilaevis*	XP_008313509.1	95.29
	*Danio rerio*	NP_001038213.2	85.48
	*Mus musculus*	NP_062760.3	71.78
	*Homo sapiens*	NP_037386.1	71.65
	*Pan troglodytes*	XP_509194.2	71.64
	*Xenopus laevis*	NP_001086516.1	64.03

## Data Availability

The data presented in this study are available in this article.

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
