# Peer review of "Molecular Identification and Expression Analysis of NOD1/2 and TBK1 in Response to Viral or Bacterial Infection in the Spotted Knifejaw (Oplegnathus punctatus)"

_animals, 2025, doi:10.3390/ani15071006_

Round 1
Reviewer 1 Report
Comments and Suggestions for Authors
The manuscript is was a pleasure to read and the research is significant to people working in the field of fish molecular biology.
- For times such as 0 h, 2 h, 3 h etc...the authors can simply use 0, 2, 3 h instead
- Italicize (in vitro)
- Figure 3 is very difficult to read
- Line 256: add a space between figure and 7
- Figure 8: consider making the y axis a maximum of 5 for all genes so it is easier to compare. opnod2 looks to be inflated but only reaches 1.4
- Lines 358-363: why were these concentrations decided?
- Line 354: why not stimulate with live or heat-killed v. harveyi?
Figure 15: should add the time of exposure/incubation with lps
- Line 399: italicize s. agalactiae
- Line 413-414: please revise, it does not make sense
Comments on the Quality of English Language
Please revise the manuscript carefully there are some grammar issues and minor formatting mistakes.
Author Response
Reviewer 1:
Comments 1: For times such as 0 h, 2 h, 3 h etc. the authors can simply use 0, 2, 3 h instead
Response 1: We thank the reviewer for this important suggestion. In response, we have simplified the description of time points accordingly.
Comments 2: Italicize (in vitro)
Response 2: We appreciate the reviewer’s comment and italicized in vitro as requested.
Comments 3: Figure 3 is very difficult to read
Response 3: We thank the reviewer for this feedback and have simplified Figure 3 to improve readability.
Comments 4: Line 256: add a space between figure and 7
Response 4 : Thank you for pointing this out. We have added the required space between "figure" and "7."
Comments 5: Figure 8: consider making the y axis a maximum of 5 for all genes so it is easier to compare. opnod2 looks to be inflated but only reaches 1.4
Response 5: We appreciate the reviewer’s suggestion and have adjusted the y-axis accordingly.
Comments 6: Lines 358-363: why were these concentrations decided?
Response 6 : We appreciate the reviewers’ inquiry regarding the rationale for selecting LPS and poly I:C concentrations in our experiments. The concentrations were determined based on a combination of literature review, preliminary dose-response experiments, and optimization for our specific experimental model (Oplegnathus punctatus).
[1] Transcriptomic profiling reveals the immune response mechanism of the Thamnaconus modestus induced by the poly (I:C) and LPS
[2] Nile Tilapia P38 Mapk Interacts with Tak1 and is Involved in the Immune Response to Streptococcus Agalactiae, Polyi:C, and Lps Stimulation.
Comments 7: Line 354: why not stimulate with live or heat-killed harveyi?
Response 7: We are grateful to the reviewer for raising this important point. Our decision to use lipopolysaccharide (LPS) instead of live or heat - killed Vibrio harveyi was made deliberately. The objective of our study was to elucidate the roles of nod1/2 and tbk1 in recognizing conserved pathogen - associated molecular patterns (PAMPs), , with a particular focus on LPS, a key component of Gram negative bacteria. Employing purified LPS enabled us to isolate and specifically investigate the immune response triggered by this PAMP. This approach helped avoid confounding factors associated with live bacteria, such as virulence factors and replication dynamics, as well as those from heat - killed bacteria, including residual metabolic byproducts
Comments 8: Figure 15: should add the time of exposure/incubation with lps
Response 8: We thank the reviewer for this helpful comment and have added the incubation time in Figure 15.
Comments 9: italicize s. agalactiae
Response 9: We sincerely apologize for this oversight and have corrected it.
Comments 10: Line 413-414: please revise, it does not make sense
Response10: We thank the reviewer for highlighting this issue and have revised the sentence.
Reviewer 2 Report
Comments and Suggestions for Authors
- Ensure consistent use of terms such as "Nod1" and "Nod2" throughout the text. For example, use "Nod1" instead of "nod1" for consistency.
- Provide more detail on the mechanisms by which Nod1 and Nod2 activate the NF-κB signaling pathway. For example: Upon recognition of bacterial components, Nod1 and Nod2 recruit and activate the receptor-interacting serine-threonine kinase 2 (RIPK2) through their CARD domains. This activation leads to the phosphorylation and activation of the IKK kinase complex, which ultimately promotes the transcription of genes involved in inflammatory responses.
- When comparing the number of LRR domains in different species, explain the significance of these differences. Example: "The variation in the number of LRR domains among different species may reflect adaptations to specific environmental pressures and pathogen exposures.
- Ensure consistent use of terminology throughout the manuscript. For example, use either "spotted knifejaw" or the scientific name consistently.
- Highlight the significance of the observed expression patterns of Opnod1, Opnod2, and Optbk1 in different tissues. Example: The high expression of Opnod1 in the skin suggests a critical role in the first line of defense against pathogens, while the expression of Opnod2 in gill and skin tissues indicates its involvement in mucosal immunity.
- Line # 71: NOD1 and NOD2 genes have also been characterized in rohu (https://doi.org/10.1016/j.dci.2011.06.018; https://doi.org/10.1016/j.fsi.2012.02.018), mrigal (https://link.springer.com/article/10.1007/s12038-013-9330-y), and catla (https://www.isca.me/IJBS/Archive/v2/i3/10.ISCA-IRJBS-2013-016.pdf). These studies could be incorporated into the discussion to provide additional context. Furthermore, the activation of NOD1 and NOD2 by bacterial infection (E. piscicida) has been investigated in https://doi.org/10.3390/vaccines11091470. Including these references may help strengthen the rationale for your hypothesis.
- Integrate more recent studies to provide a broader context for your research. Discuss how your findings align with or differ from previous studies.
Comments on the Quality of English Language
Need to be improved
Author Response
Reviewer 2:
Comments 1: Ensure consistent use of terms such as "Nod1" and "Nod2" throughout the text. For example, use "Nod1" instead of "nod1" for consistency.
Response 1: We appreciate the reminder. For gene names, we have followed standard conventions by lowercase and italics (e.g., nod1 and nod2). For protein names, we have used uppercase and regular font (e.g., NOD1 and NOD2) for consistency throughout the manuscript.
Comments 2: Provide more detail on the mechanisms by which Nod1 and Nod2 activate the NF-κB signaling pathway. For example: Upon recognition of bacterial components, Nod1 and Nod2 recruit and activate the receptor-interacting serine-threonine kinase 2 (RIPK2) through their CARD domains. This activation leads to the phosphorylation and activation of the IKK kinase complex, which ultimately promotes the transcription of genes involved in inflammatory responses.
Response 2: We thank the reviewer for this valuable suggestion. We have expanded the description of the mechanisms by which Nod1 and Nod2 activate the NF-κB signaling pathway.
Comments 3: When comparing the number of LRR domains in different species, explain the significance of these differences. Example: "The variation in the number of LRR domains among different species may reflect adaptations to specific environmental pressures and pathogen exposures.
Response 3: We thank the reviewer for this insight and have elaborated on the significance of these differences. We have added two references.
- Li Z, Shang D. NOD1 and NOD2: Essential Monitoring Partners in the Innate Immune System. CURR ISSUES MOL BIOL. 2024 46:9463-79.
- Liao Z, Su J. Progresses on three pattern recognition receptor families (TLRs, RLRs and NLRs) in teleost. DEV COMP IMMUNOL. 2021 122:104131.
Comments 4: Highlight the significance of the observed expression patterns of Opnod1, Opnod2, and Optbk1 in different tissues. Example: The high expression of Opnod1 in the skin suggests a critical role in the first line of defense against pathogens, while the expression of Opnod2 in gill and skin tissues indicates its involvement in mucosal immunity.
Response 4: We greatly appreciate this suggestion. We have revised the manuscript to emphasize the significance of the expression patterns of Opnod1, Opnod2, and Optbk1.
Comments 5: Line # 71: NOD1 and NOD2 genes have also been characterized in rohu (https://doi.org/10.1016/j.dci.2011.06.018; https://doi.org/10.1016/j.fsi.2012.02.018), mrigal (https://link.springer.com/article/10.1007/s12038-013-9330-y), and catla (https://www.isca.me/IJBS/Archive/v2/i3/10.ISCA-IRJBS-2013-016.pdf). These studies could be incorporated into the discussion to provide additional context. Furthermore, the activation of NOD1 and NOD2 by bacterial infection (E. piscicida) has been investigated in https://doi.org/10.3390/vaccines11091470. Including these references may help strengthen the rationale for your hypothesis.
Response 5: We agree this is an excellent suggestion. As recommended, we have incorporated additional references to support our hypothesis.
1.Mrigal (Cirrhinus mrigala):NOD1 and NOD2 receptors in mrigal (Cirrhinus mrigala): Inductive expression and downstream signaling in ligand stimulation and bacterial infections
2.Rohu (Labeo rohita):Molecular cloning and characterization of nucleotide binding and oligomerization domain-1 (NOD1) receptor in the Indian Major Carp, rohu (Labeo rohita), and analysis of its inductive expression and down-stream signalling molecules
following ligands exposure and Gram-negative bacterial infections
3.Catla (Catla catla):Nucleotide Binding and Oligomerization Domain 1 (NOD1) Receptor in Catla (Catla catla): Inductive Expression and Down-Stream Signaling in
Ligand Stimulation and Bacterial Infections
4. Activation of NOD1 and NOD2 by bacterial infection:Recombinant Attenuated Edwardsiella piscicida Vaccine Displaying Regulated Lysis to Confer Biological Containment and Protect Catfish against Edwardsiellosis.
Comments 6: Integrate more recent studies to provide a broader context for your research. Discuss how your findings align with or differ from previous studies.
Response 6: We thank the reviewer for this comment. We have cited more recent studies to support our conclusions.
Reviewer 3 Report
Comments and Suggestions for Authors
This manuscript investigated the roles of the Opnod1, Opnod2, and Optbk1 genes in both antiviral and antibacterial immunity in the spotted knifejaw. The authors also used qRT-PCR to validate the expression patterns of Opnod1, Opnod2, and Optbk1 mRNA in different tissues and at various time points. While the work is interesting, some issues such as the experimental methods and formatting of this manuscript need to be further supplemented and improved.
Comments:
1.Page 1, line 4: Oplegnathus punctatus should be italicized.
2.Line 43: Gene names should be in lower case and italicized. Please revise accordingly. There are similar problems throughout the manuscript, please revise them one by one.
3.Page 8, line 216: Figure 3A is blurry, please adjust it.
4.Pages 11, 12, and 13, lines 257, 260, and 263: The images are not clear enough, please adjust them.
5.Page 15, line 279: Figure 8 and subsequent images are somewhat too small, and the fonts are inconsistent, with some being blurry. Please revise them uniformly.
6.Page 22, line 458: The formatting in the References section is inconsistent.
7.Line 124: Is the challenge concentration of 10^9 too high? Based on experience, it is highly likely that the subjects may not survive until the seventh day.
8.The background information on in vitro stimulation with poly I:C and LPS in the introduction is somewhat lacking.
9.Figure 9: The expression level of Opnod1 in the kidney at 0 days (pre-challenge) does not match its corresponding expression level in the kidney shown in Figure 8.
Author Response
Reviewer 3:
Comments 1: Page 1, line 4: Oplegnathus punctatus should be italicized.
Response 1: Thank you for your careful review. We apologize for the oversight and have corrected it.
Comments 2: Line 43: Gene names should be in lower case and italicized. Please revise accordingly. There are similar problems throughout the manuscript, please revise them one by one.
Response 2: We sincerely apologize for this error. We have carefully revised all gene names to ensure consistency, following the proper formatting with lowercase and italics.
Comments 3: Page 8, line 216: Figure 3A is blurry, please adjust it.
Response 3: We appreciate your feedback and have replaced the figure with a higher-resolution version to improve clarity.
Comments 4: Pages 11, 12, and 13, lines 257, 260, and 263: The images are not clear enough, please adjust them.
Response 4: Thank you for this suggestion. We have improved the resolution of these figures, including Figure 8 and the subsequent images, to improve clarity and consistency.
Comments 5: Page 15, line 279: Figure 8 and subsequent images are somewhat too small, and the fonts are inconsistent, with some being blurry. Please revise them uniformly.
Response 5: We apologize for this oversight and have resized and standardized the fonts in all figures, including Figure 8 and the subsequent images, to improve clarity and consistency.
Comments 6: Page 22, line 458: The formatting in the References section is inconsistent.
Response 6: We apologize for the formatting issues and have revised the References section to ensure consistency.
Comments 7: Line 124: Is the challenge concentration of 10^9 too high? Based on experience, it is highly likely that the subjects may not survive until the seventh day.
Response 7: We appreciate your concern. The concentration of 10^9 was determined through preliminary experiments, and based on these testes, the subjects survived until the seventh day.
Comments 8: The background information on in vitro stimulation with poly I:C and LPS in the introduction is somewhat lacking.
Response 8: We thank the reviewer for this suggestion. We have added more details on poly I:C and LPS stimulation in the Introduction to provide a clearer background for the study. Additionally, we have included the following references to support this section.
[35] Xu A, Han F, Zhang Y, Chen S, Bian L, Gao T. Transcriptomic profiling reveals the immune response mechanism of the Thamnaconus modestus induced by the poly (I:C) and LPS. GENE. 2024 897:148065.
[36] Hou QH, Yi SB, Ding X, Zhang HX, Sun Y, Zhang Y, et al. Differential expression analysis of nuclear oligomerization domain proteins NOD1 and NOD2 in orange-spotted grouper (Epinephelus coioides). Fish Shellfish Immunol. 2012 33:1102-11.
Comments 9: Figure 9: The expression level of Opnod1 in the kidney at 0 days (pre-challenge) does not match its corresponding expression level in the kidney shown in Figure 8.
Response 9: We thank the reviewer for highlighting this discrepancy. The difference arises from the use of different controls in the two figures.
Round 2
Reviewer 2 Report
Comments and Suggestions for Authors
Authors have successfully revised the manuscript.
Author Response
Dear Reviewer,
Thank you very much for your advise.